# Parallel Correlation Clustering on Big Graphs

**Xinghao Pan**[α,ε], **Dimitris Papailiopoulos**[α,ε], **Samet Oymak**[α,ε],
**Benjamin Recht**[α,ε,σ], **Kannan Ramchandran**[ε], **and Michael I. Jordan**[α,ε,σ]
[α]AMPLab, [ε]EECS at UC Berkeley, [σ]Statistics at UC Berkeley

## Abstract

Given a similarity graph between items, correlation clustering (CC) groups similar items together and dissimilar ones apart. One of the most popular CC algorithms is *KwikCluster*: an algorithm that serially clusters neighborhoods of vertices, and obtains a 3-approximation ratio. Unfortunately, in practice *KwikCluster* requires a large number of clustering rounds, a potential bottleneck for large graphs.

We present *C4* and *ClusterWild!*, two algorithms for parallel correlation clustering that run in a polylogarithmic number of rounds, and provably achieve nearly linear speedups. *C4* uses concurrency control to enforce serializability of a parallel clustering process, and guarantees a 3-approximation ratio. *ClusterWild!* is a coordination free algorithm that abandons consistency for the benefit of better scaling; this leads to a provably small loss in the 3 approximation ratio.

We demonstrate experimentally that both algorithms outperform the state of the art, both in terms of clustering accuracy and running time. We show that our algorithms can cluster billion-edge graphs in under 5 seconds on 32 cores, while achieving a $15\times$ speedup.

## 1 Introduction

Clustering items according to some notion of similarity is a major primitive in machine learning. *Correlation clustering* serves as a basic means to achieve this goal: given a similarity measure between items, the goal is to group similar items together and dissimilar items apart. In contrast to other clustering approaches, the number of clusters is not determined a priori, and good solutions aim to balance the tension between grouping all items together versus isolating them.

The simplest CC variant can be described on a complete signed graph. Our input is a graph $G$ on $n$ vertices, with $+1$ weights on edges between similar items, and $-1$ edges between dissimilar ones. Our goal is to generate a partition of vertices into disjoint sets that minimizes the number of *disagreeing edges*: this equals the number of "+" edges cut by the clusters plus the number of "−" edges inside the clusters. This metric is commonly called the *number of disagreements*. In Figure 1, we give a toy example of a CC instance.

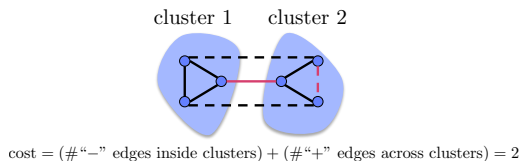

cost = (#"−" edges inside clusters) + (#"+" edges across clusters) = 2

Figure 1: In the above graph, solid edges denote similarity and dashed dissimilarity. The number of disagreeing edges in the above clustering clustering is 2; we color the bad edges with red.

Entity deduplication is the archetypal motivating example for correlation clustering, with applications in chat disentanglement, co-reference resolution, and spam detection [1, 2, 3, 4, 5, 6]. The input is a set of entities (say, results of a keyword search), and a pairwise classifier that indicates—with some error—similarities between entities. Two results of a keyword search might refer to the same item, but might look different if they come from different sources. By building a similarity

graph between entities and then applying CC, the hope is to cluster duplicate entities in the same group; in the context of keyword search, this implies a more meaningful and compact list of results. CC has been further applied to finding communities in signed networks, classifying missing edges in opinion or trust networks [7, 8], gene clustering [9], and consensus clustering [3].

*KwikCluster* is the simplest CC algorithm that achieves a provable 3-approximation ratio [10], and works in the following way: pick a vertex $v$ at random (a *cluster center*), create a cluster for $v$ and its positive neighborhood $N(v)$ (i.e., vertices connected to $v$ with positive edges), peel these vertices and their associated edges from the graph, and repeat until all vertices are clustered. Beyond its theoretical guarantees, experimentally *KwikCluster* performs well when combined with local heuristics [3].

*KwikCluster* seems like an inherently sequential algorithm, and in most cases of interest it requires many peeling rounds. This happens because a small number of vertices are clustered per round. This can be a bottleneck for large graphs. Recently, there have been efforts to develop scalable variants of *KwikCluster* [5, 6]. In [6] a distributed peeling algorithm was presented in the context of MapReduce. Using an elegant analysis, the authors establish a $(3 + \epsilon)$-approximation in a polylogarithmic number of rounds. The algorithm employs a simple step that rejects vertices that are executed in parallel but are "conflicting"; however, we see in our experiments, this seemingly minor coordination step hinders scale-ups in a parallel core setting. In [5], a sketch of a distributed algorithm was presented. This algorithm achieves the same approximation as *KwikCluster*, in a logarithmic number of rounds, in expectation. However, it performs significant redundant work, per iteration, in its effort to detect in parallel which vertices should become cluster centers.

**Our contributions**  We present *C4* and *ClusterWild!*, two parallel CC algorithms with provable performance guarantees, that in practice outperform the state of the art, both in terms of running time and clustering accuracy. *C4* is a parallel version of *KwikCluster* that uses concurrency control to establish a 3-approximation ratio. *ClusterWild!* is a simple to implement, coordination-free algorithm that abandons consistency for the benefit of better scaling, while having a provably small loss in the 3 approximation ratio.

*C4* achieves a 3 approximation ratio, in a poly-logarithmic number of rounds, by enforcing consistency between concurrently running peeling threads. Consistency is enforced using *concurrency control*, a notion extensively studied for databases transactions, that was recently used to parallelize inherently sequential machine learning algorithms [11].

*ClusterWild!* is a coordination-free parallel CC algorithm that waives consistency in favor of speed. The cost we pay is an arbitrarily small loss in *ClusterWild!*'s accuracy. We show that *ClusterWild!* achieves a $(3 + \epsilon)\mathsf{OPT} + O(\epsilon \cdot n \cdot \log^2 n)$ approximation, in a poly-logarithmic number of rounds, with provable nearly linear speedups. Our main theoretical innovation for *ClusterWild!* is analyzing the coordination-free algorithm as a serial variant of *KwikCluster* that runs on a "noisy" graph.

In our experimental evaluation, we demonstrate that both algorithms gracefully scale up to graphs with billions of edges. In these large graphs, our algorithms output a valid clustering in less than 5 seconds, on 32 threads, up to an order of magnitude faster than *KwikCluster*. We observe how, not unexpectedly, *ClusterWild!* is faster than *C4*, and quite surprisingly, abandoning coordination in this parallel setting, only amounts to a 1% of relative loss in the clustering accuracy. Furthermore, we compare against state of the art parallel CC algorithms, showing that we consistently outperform these algorithms in terms of both running time and clustering accuracy.

**Notation**  $G$ denotes a graph with $n$ vertices and $m$ edges. $G$ is complete and only has $\pm 1$ edges. We denote by $d_v$ the positive degree of a vertex, i.e., the number of vertices connected to $v$ with positive edges. $\Delta$ denotes the positive maximum degree of $G$, and $N(v)$ denotes the positive neighborhood of $v$; moreover, let $C_v = \{v, N(v)\}$. Two vertices $u, v$ are termed as "friends" if $u \in N(v)$ and vice versa. We denote by $\pi$ a permutation of $\{1, \ldots, n\}$.

## 2  Two Parallel Algorithms for Correlation Clustering

The formal definition of correlation clustering is given below.

**Correlation Clustering.** *Given a graph $G$ on $n$ vertices, partition the vertices into an arbitrary number $k$ of disjoint subsets $\mathcal{C}_1, \ldots, \mathcal{C}_k$ such that the sum of negative edges within the subsets plus the sum of positive edges across the subsets is minimized:*

$$\mathsf{OPT} = \min_{1 \leq k \leq n} \min_{\substack{\mathcal{C}_i \cap \mathcal{C}_j = 0, \forall i \neq j \\ \cup_{i=1}^{k} \mathcal{C}_i = \{1, \ldots, n\}}} \sum_{i=1}^{k} E^-(\mathcal{C}_i, \mathcal{C}_i) + \sum_{i=1}^{k} \sum_{j=i+1}^{k} E^+(\mathcal{C}_i, \mathcal{C}_j)$$

*where $E^+$ and $E^-$ are the sets of positive and negative edges in G.*

*KwikCluster* is a remarkably simple algorithm that approximately solves the above combinatorial problem, and operates as follows. A random vertex $v$ is picked, a cluster $C_v$ is created with $v$ and its positive neighborhood, then the vertices in $C_v$ are peeled from the graph, and this process is repeated until all vertices are clustered *KwikCluster* can be equivalently executed, as noted by [5], if we substitute the random choice of a vertex per peeling round, with a random order $\pi$ preassigned to vertices, (see Alg. 1). That is, select a random permutation on vertices, then peel the vertex indexed by $\pi(1)$, and its friends. Remove from $\pi$ the vertices in $C_v$ and repeat this process. Having an order among vertices makes the discussion of parallel algorithms more convenient.

*C4*: **Parallel CC using Concurency Control.** Suppose we now wish to run a parallel version of *KwikCluster*, say on two threads: one thread picks vertex $v$ indexed by $\pi(1)$ and the other thread picks $u$ indexed by $\pi(2)$, concurrently. Can both vertices be cluster centers? They can, iff they are not friends in $G$. If $v$ and $u$ are connected with a positive edge, then the vertex with the smallest order wins. This is our *concurency*

---
**Algorithm 1** *KwikCluster* with $\pi$

---
1: $\pi$ = a random permutation of $\{1, \ldots, n\}$
2: **while** $V \neq \emptyset$ **do**
3:    select the vertex $v$ indexed by $\pi(1)$
4:    $C_v = \{v, N(v)\}$
5:    Remove clustered vertices from $G$ and $\pi$
6: **end while**

---

*rule no. 1.* Now, assume that $v$ and $u$ are not friends in $G$, and both $v$ and $u$ become cluster centers. Moreover, assume that $v$ and $u$ have a common, unclustered friend, say $w$: should $w$ be clustered with $v$, or $u$? We need to follow what would happen with *KwikCluster* in Alg. 1: $w$ will go with the vertex that has the smallest permutation number, in this case $v$. This is *concurrency rule no. 2*. Following the above simple rules, we develop *C4*, our serializable parallel CC algorithm. Since, *C4* constructs the same clusters as *KwikCluster* (for a given ordering $\pi$), it inherits its 3 approximation. The above idea of identifying the cluster centers in rounds was first used in [12] to obtain a parallel algorithm for maximal independent set (MIS).

*C4*, shown as Alg. 2, starts by assigning a random permutation $\pi$ to the vertices, it then samples an active set $\mathcal{A}$ of $\frac{n}{\Delta}$ unclustered vertices; this sample is taken from the prefix of $\pi$. After sampling $\mathcal{A}$, each of the $P$ threads picks a vertex with the smallest order in $\mathcal{A}$, then checks if that vertex can become a cluster center. We first enforce *concurrency rule no. 1*: adjacent vertices cannot be cluster centers at the same time. *C4* enforces it by making each thread check the friends of the vertex, say $v$, that is picked from $\mathcal{A}$. A thread will check in `attemptCluster` whether its vertex $v$ has any preceding friends that are cluster centers. If there are none, it will go ahead and label $v$ as cluster center, and proceed with creating a cluster. If a preceding friend of $v$ is a cluster center, then $v$ is labeled as not being a cluster center. If a preceding friend of $v$, call it $u$, has not yet received a label (i.e., $u$ is currently being processed and is not yet labeled as cluster center or not), then the thread processing $v$, will wait on $u$ to receive a label. The major technical detail is in showing that this wait time is bounded; we show that no more than $O(\log n)$ threads can be in conflict at the same time, using a new subgraph sampling lemma [13]. Since *C4* is serializable, it has to respect *concurrency rule no. 2*: if a vertex $u$ is adjacency to two cluster centers, then it gets assigned to the one with smaller permutation order. This is accomplished in `createCluster`. After processing all vertices in $\mathcal{A}$, all threads are synchronized in bulk, the clustered vertices are removed, a new active set is sampled, and the same process is repeated until everything has been clustered. In the following section, we present the theoretical guarantees for *C4*.

**Algorithm 2** *C4 & ClusterWild!*

1: **Input**: $G, \epsilon$
2: clusterID$(1) = \ldots =$ clusterID$(n) = \infty$
3: $\pi$ = a random permutation of $\{1, \ldots, n\}$
4: **while** $V \neq \emptyset$ **do**
5: $\quad \Delta =$ maximum vertex degree in $G(V)$
6: $\quad \mathcal{A} =$ the first $\epsilon \cdot \frac{n}{\Delta}$ vertices in $V[\pi]$.
7: $\quad$ **while** $\mathcal{A} \neq \emptyset$ **do in parallel**
8: $\quad\quad v =$ first element in $\mathcal{A}$
9: $\quad\quad \mathcal{A} = \mathcal{A} - \{v\}$
10: $\quad\quad$ **if** *C4* **then** // concurrency control
11: $\quad\quad\quad$ `attemptCluster`$(v)$
12: $\quad\quad$ **else if** *ClusterWild!* **then** // coordination free
13: $\quad\quad\quad$ `createCluster`$(v)$
14: $\quad\quad$ **end if**
15: $\quad$ **end while**
16: $\quad$ Remove clustered vertices from $V$ and $\pi$
17: **end while**
18: **Output:** $\{\text{clusterID}(1), \ldots, \text{clusterID}(n)\}$.

`createCluster`$(v)$:
$\quad$ clusterID$(v) = \pi(v)$
$\quad$ **for** $u \in \Gamma(v) \setminus \mathcal{A}$ **do**
$\quad\quad$ clusterID$(u) = \min(\text{clusterID}(u), \pi(v))$
$\quad$ **end for**

`attemptCluster`$(v)$:
$\quad$ **if** clusterID$(u) = \infty$ **and** `isCenter`$(v)$ **then**
$\quad\quad$ `createCluster`$(v)$
$\quad$ **end if**

`isCenter`$(v)$:
$\quad$ **for** $u \in \Gamma(v)$ **do** // check friends (in order of $\pi$)
$\quad\quad$ **if** $\pi(u) < \pi(v)$ **then** // if they precede you, wait
$\quad\quad\quad$ **wait** until clusterID$(u) \neq \infty$ // till clustered
$\quad\quad\quad$ **if** `isCenter`$(u)$ **then**
$\quad\quad\quad\quad$ **return** 0 //a friend is center, so you can't be
$\quad\quad\quad$ **end if**
$\quad\quad$ **end if**
$\quad$ **end for**
$\quad$ **return** 1 // no earlier friends are centers, so you are

*ClusterWild!*: **Coordination-free Correlation Clustering.** *ClusterWild!* speeds up computation by ignoring the first concurrency rule. It uniformly samples unclustered vertices, and builds clusters around *all of them*, without respecting the rule that cluster centers cannot be friends in $G$. In *ClusterWild!*, threads bypass the `attemptCluster` routine; this eliminates the "waiting" part of *C4*. *ClusterWild!* samples a set $\mathcal{A}$ of vertices from the prefix of $\pi$. Each thread picks the first ordered vertex remaining in $\mathcal{A}$, and using that vertex as a cluster center, it creates a cluster around it. It peels away the clustered vertices and repeats the same process, on the next remaining vertex in $\mathcal{A}$. At the end of processing all vertices in $\mathcal{A}$, all threads are synchronized in bulk, the clustered vertices are removed, a new active set is sampled, and the parallel clustering is repeated. A careful analysis along the lines of [6] shows that the number of rounds (i.e., bulk synchronization steps) is only poly-logarithmic.

Quite unsurprisingly, *ClusterWild!* is faster than *C4*. Interestingly, abandoning consistency does not incur much loss in the approximation ratio. We show how the error introduced in the accuracy of the solution can be bounded. We characterize this error theoretically, and show that in practice it only translates to only a relative 1% loss in the objective. The main intuition of why *ClusterWild!* does not introduce too much error is that the chance of two randomly selected vertices being friends in $G$ is small, hence the concurrency rules are infrequently broken.

## 3 Theoretical Guarantees

In this section, we bound the number of rounds required for each algorithms, and establish the theoretical speedup one can obtain with $P$ parallel threads. We proceed to present our approximation guarantees. We would like to remind the reader that—as in relevant literature—we consider graphs that are complete, signed, and unweighted. The omitted proofs can be found in the Appendix.

### 3.1 Number of rounds and running time

Our analysis follows those of [12] and [6]. The main idea is to track how fast the maximum degree decreases in the remaining graph at the end of each round.

**Lemma 1.** *C4 and ClusterWild! terminate after* $O\left(\frac{1}{\epsilon} \log n \cdot \log \Delta\right)$ *rounds w.h.p.*

We now analyze the running time of both algorithms under a simplified BSP model. The main idea is that the the running time of each "super step" (i.e., round) is determined by the "straggling" thread (i.e., the one that gets assigned the most amount of work), plus the time needed for synchronization at the end of each round.

**Assumption 1.** *We assume that threads operate asynchronously within a round and synchronize at the end of a round. A memory cell can be written/read concurrently by multiple threads. The time*

*spent per round of the algorithm is proportional to the time of the slowest thread. The cost of thread synchronization at the end of each batch takes time $O(P)$, where $P$ is the number of threads. The total computation cost is proportional to the sum of the time spent for all rounds, plus the time spent during the bulk synchronization step.*

Under this simplified model, we show that both algorithms obtain nearly linear speedup, with *ClusterWild!* being faster than *C4*, precisely due to lack of coordination. Our main tool for analyzing *C4* is a recent graph-theoretic result from [13] (Theorem 1), which guarantees that if one samples an $O(n/\Delta)$ subset of vertices in a graph, the sampled subgraph has a connected component of size at most $O(\log n)$. Combining the above, in the appendix we show the following result.

**Theorem 2.** *The theoretical running time of C4 on $P$ cores is upper bounded by $O\left(\left(\frac{m+n\log n}{P} + P\right)\log n \cdot \log \Delta\right)$ as long as the number of cores $P$ is smaller than $\min_i \frac{n_i}{\Delta_i}$, where $\frac{n_i}{\Delta_i}$ is the size of the batch in the $i$-th round of each algorithm. The running time of Cluster-Wild! on $P$ cores is upper bounded by $O\left(\left(\frac{m+n}{P} + P\right)\log n \cdot \log \Delta\right)$.*

### 3.2 Approximation ratio

We now proceed with establishing the approximation ratios of *C4* and *ClusterWild!*.

*C4* **is serializable.** It is straightforward that *C4* obtains precisely the same approximation ratio as *KwikCluster*. One has to simply show that for any permutation $\pi$, *KwikCluster* and *C4* will output the same clustering. This is indeed true, as the two simple concurrency rules mentioned in the previous section are sufficient for *C4* to be equivalent to *KwikCluster*.

**Theorem 3.** *C4 achieves a $3$ approximation ratio, in expectation.*

*ClusterWild!* **as a serial procedure on a noisy graph.** Analyzing *ClusterWild!* is a bit more involved. Our guarantees are based on the fact that *ClusterWild!* can be treated *as if* one was running a peeling algorithm on a "noisy" graph. Since adjacent active vertices can still become cluster centers in *ClusterWild!*, one can view the edges between them as "deleted," by a somewhat unconventional adversary. We analyze this new, noisy graph and establish our theoretical result.

**Theorem 4.** *ClusterWild! achieves a $(3+\epsilon)\cdot\mathsf{OPT}+O(\epsilon\cdot n\cdot\log^2 n)$ approximation, in expectation.*

We provide a sketch of the proof, and delegate the details to the appendix. Since *ClusterWild!* ignores the edges among active vertices, we treat these edges as deleted. In our main result, we quantify the loss of clustering accuracy that is caused by ignoring these edges. Before we proceed, we define *bad triangles*, a combinatorial structure that is used to measure the clustering quality of a peeling algorithm.

**Definition 1.** *A bad triangle in $G$ is a set of three vertices, such that two pairs are joined with a positive edge, and one pair is joined with a negative edge. Let $\mathcal{T}_b$ denote the set of bad triangles in $G$.*

To quantify the cost of *ClusterWild!*, we make the below observation.

**Lemma 5.** *The cost of any greedy algorithm that picks a vertex $v$ (irrespective of the sampling order), creates $\mathcal{C}_v$, peels it away and repeats, is equal to the number of bad triangles adjacent to each cluster center $v$.*

**Lemma 6.** *Let $\hat{G}$ denote the random graph induced by deleting all edges between active vertices per round, for a given run of ClusterWild!, and let $\tau_{new}$ denote the number of additional bad triangles that $\hat{G}$ has compared to $G$. Then, the expected cost of ClusterWild! can be upper bounded as $\mathbb{E}\left\{\sum_{t\in\mathcal{T}_b}\mathbf{1}_{\mathcal{P}_t}+\tau_{new}\right\}$, where $\mathcal{P}_t$ is the event that triangle $t$, with end points $i, j, k$, is bad, and at least one of its end points becomes active, while $t$ is still part of the original unclustered graph.*

*Proof.* We begin by bounding the second term $\mathbb{E}\{\tau_{\text{new}}\}$, by considering the number of new bad triangles $\tau_{\text{new},i}$ created at each round $i$:

$$\mathbb{E}\left\{\tau_{\text{new},i}\right\} \leq \sum_{(u,v)\in E}\mathbb{P}(u,v\in\mathcal{A}_i)\cdot|N(u)\cup N(v)| \leq \sum_{(u,v)\in E}\left(\frac{\epsilon}{\Delta_i}\right)^2\cdot 2\cdot\Delta_i \leq 2\cdot\epsilon^2\cdot\frac{E}{\Delta_i} \leq 2\cdot\epsilon^2\cdot n.$$

Using the result that *ClusterWild!* terminates after at most $O(\frac{1}{\epsilon} \log n \log \Delta)$ rounds, we get that[1] $\mathbb{E}\{\tau_{\text{new}}\} \leq O(\epsilon \cdot n \cdot \log^2 n)$.

We are left to bound $\mathbb{E}\left\{\sum_{t \in \mathcal{T}_b} \mathbf{1}_{\mathcal{P}_t}\right\} = \sum_{t \in \mathcal{T}_b} p_t$. To do that we use the following lemma.

**Lemma 7.** *If $p_t$ satisfies $\forall e$, $\sum_{t:e \subset t \in \mathcal{T}_b} \frac{p_t}{\alpha} \leq 1$, then, $\sum_{t \in \mathcal{T}_b} p_t \leq \alpha \cdot OPT$.*

*Proof.* Let $\mathcal{B}_*$ be one (of the possibly many) sets of edges that attribute a $+1$ in the cost of an optimal algorithm. Then, $\text{OPT} = \sum_{e \in \mathcal{B}^*} 1 \geq \sum_{e \in \mathcal{B}^*} \sum_{t:e \subset t \in \mathcal{T}_b} \frac{p_t}{\alpha} = \sum_{t \in \mathcal{T}_b} \underbrace{|B_* \cap t|}_{\geq 1} \frac{p_t}{\alpha} \geq \sum_{t \in \mathcal{T}_b} \frac{p_t}{\alpha}$. $\square$

Now, as with [6], we will simply have to bound the expectation of the bad triangles, adjacent to an edge $(u,v)$: $\sum_{t:\{u,v\} \subset t \in \mathcal{T}_b} \mathbf{1}_{\mathcal{P}_t}$. Let $\mathcal{S}_{u,v} = \bigcup_{\{u,v\} \subset t \in \mathcal{T}_b} t$ be the union of the sets of nodes of the bad triangles that contain both vertices $u$ and $v$. Observe that if some $w \in S \backslash \{u,v\}$ becomes active before $u$ and $v$, then a cost of 1 (i.e., the cost of the bad triangle $\{u,v,w\}$) is incurred. On the other hand, if either $u$ or $v$, or both, are selected as pivots in some round, then $\mathcal{C}_{u,v}$ can be as high as $|S| - 2$, i.e., at most equal to all bad triangles containing the edge $\{u,v\}$. Let $A_{uv} = \{u \text{ or } v \text{ are activated before any other vertices in } S_{u,v}\}$. Then,

$$\mathbb{E}\left[\mathcal{C}_{u,v}\right] = \mathbb{E}\left[\mathcal{C}_{u,v} | A_{u,v}\right] \cdot \mathbb{P}(A_{u,v}) + \mathbb{E}\left[\mathcal{C}_{u,v} | A_{u,v}^C\right] \cdot \mathbb{P}(A_{u,v}^C)$$
$$\leq 1 + (|S| - 2) \cdot \mathbb{P}(\{u,v\} \cap \mathcal{A} \neq \emptyset | \mathcal{S} \cap \mathcal{A} \neq \emptyset) \leq 1 + 2|S| \cdot \mathbb{P}(v \cap \mathcal{A} \neq \emptyset | \mathcal{S} \cap \mathcal{A} \neq \emptyset)$$

where the last inequality is obtained by a union bound over $u$ and $v$. We now bound the following probability:

$$\mathbb{P}\{v \in \mathcal{A} | \mathcal{S} \cap \mathcal{A} \neq \emptyset\} = \frac{\mathbb{P}\{v \in \mathcal{A}\} \cdot \mathbb{P}\{\mathcal{S} \cap \mathcal{A} \neq \emptyset | v \in \mathcal{A}\}}{\mathbb{P}\{\mathcal{S} \cap \mathcal{A} \neq \emptyset\}} = \frac{\mathbb{P}\{v \in \mathcal{A}\}}{\mathbb{P}\{\mathcal{S} \cap \mathcal{A} \neq \emptyset\}} = \frac{\mathbb{P}\{v \in \mathcal{A}\}}{1 - \mathbb{P}\{\mathcal{S} \cap \mathcal{A} = \emptyset\}}.$$

Observe that $\mathbb{P}\{v \in \mathcal{A}\} = \frac{\epsilon}{\Delta}$, hence we need to upper bound $\mathbb{P}\{\mathcal{S} \cap \mathcal{A} = \emptyset\}$. The probability, per round, that no positive neighbors in $\mathcal{S}$ become activated is upper bounded by

$$\frac{\binom{n-|\mathcal{S}|}{P}}{\binom{n}{P}} = \prod_{t=1}^{|\mathcal{S}|}\left(1 - \frac{P}{n - |\mathcal{S}| + t}\right) \leq \left(1 - \frac{P}{n}\right)^{|\mathcal{S}|} = \left[\left(1 - \frac{P}{n}\right)^{n/P}\right]^{|\mathcal{S}|n/P} \leq \left(\frac{1}{e}\right)^{|\mathcal{S}|n/P}.$$

Hence, the probability can be upper bounded as

$$|\mathcal{S}|\mathbb{P}\{v \cap \mathcal{A} \neq \emptyset | \mathcal{S} \cap \mathcal{A} \neq \emptyset\} \leq \frac{\epsilon \cdot |\mathcal{S}|/\Delta}{1 - e^{-\epsilon \cdot |\mathcal{S}|/\Delta}}.$$

We know that $|\mathcal{S}| \leq 2 \cdot \Delta + 2$ and also $\epsilon \leq 1$. Then, $0 \leq \epsilon \cdot \frac{|\mathcal{S}|}{\Delta} \leq \epsilon \cdot \left(2 + \frac{2}{\Delta}\right) \leq 4$ Hence, we have $\mathbb{E}(\mathcal{C}_{u,v}) \leq 1 + 2 \cdot \frac{4\epsilon}{1 - \exp\{-4\epsilon\}}$. The overall expectation is then bounded by $\mathbb{E}\left\{\sum_{t \in \mathcal{T}_b} \mathbf{1}_{\mathcal{P}_t} + \tau_{\text{new}}\right\} \leq \left(1 + 2 \cdot \frac{4 \cdot \epsilon}{1 - e^{-4 \cdot \epsilon}}\right) \cdot \text{OPT} + \epsilon \cdot n \cdot \ln(n/\delta) \cdot \log \Delta \leq (3 + \epsilon) \cdot \text{OPT} + O(\epsilon \cdot n \cdot \log^2 n)$ which establishes our approximation ratio for *ClusterWild!*. $\square$

### 3.3 BSP Algorithms as a Proxy for Asynchronous Algorithms

We would like to note that the analysis under the BSP model can be a useful proxy for the performance of completely asynchronous variants of our algorithms. Specifically, see Alg. 3, where we remove the synchronization barriers.

The only difference between the asynchronous execution in Alg. 3, compared to Alg. 2, is the complete lack of bulk synchronization, at the end of the processing of each active set $\mathcal{A}$. Although the analysis of the BSP variants of the algorithms is tractable, unfortunately analyzing precisely the speedup of the asynchronous *C4* and the approximation guarantees for the asynchronous *ClusterWild!* is challenging. However, in our experimental section we test the completely asynchronous algorithms against the BSP algorithms of the previous section, and observe that they perform quite similarly both in terms of accuracy of clustering, and running times.

## 4 Related Work

Correlation clustering was formally introduced by Bansal et al. [14]. In the general case, minimizing disagreements is NP-hard and hard to approximate within an arbitrarily small constant (APX-hard) [14, 15]. There are two variations of the problem: *i)* CC on complete graphs where all edges are present and all weights are $\pm 1$, and *ii)* CC on general graphs with arbitrary edge weights. Both problems are hard, however the general graph setup seems fundamentally harder. The best known approximation ratio for the latter is $O(\log n)$, and a reduction to the minimum multicut problem indicates that any improvement to that requires fundamental breakthroughs in theoretical algorithms [16].

---

**Algorithm 3** *C4 & ClusterWild!*
(asynchronous execution)

1: **Input**: $G$
2: clusterID(1) = ... = clusterID($n$) = $\infty$
3: $\pi$ = a random permutation of $\{1, \ldots, n\}$
4: **while** $V \neq \emptyset$ **do**
5:    $v$ = first element in $V$
6:    $V = V - \{v\}$
7:    **if** *C4* **then** // concurrency control
8:       attemptCluster($v$)
9:    **else if** *ClusterWild!* **then** // coordination free
10:       createCluster($v$)
11:    **end if**
12:    Remove clustered vertices from $V$ and $\pi$
13: **end while**
14: **Output**: $\{\text{clusterID}(1), \ldots, \text{clusterID}(n)\}$.

---

In the case of complete unweighted graphs, a long series of results establishes a 2.5 approximation via a rounded linear program (LP) [10]. A recent result establishes a 2.06 approximation using an elegant rounding to the same LP relaxation [17]. By avoiding the expensive LP, and by just using the rounding procedure of [10] as a basis for a greedy algorithm yields *KwikCluster*: a 3 approximation for CC on complete unweighted graphs.

Variations of the cost metric for CC change the algorithmic landscape: maximizing agreements (the dual measure of disagreements) [14, 18, 19], or maximizing the difference between the number of agreements and disagreements [20, 21], come with different hardness and approximation results. There are also several variants: chromatic CC [22], overlapping CC [23], small number of clusters CC with added constraints that are suitable for some biology applications [24].

The way *C4* finds the cluster centers can be seen as a variation of the MIS algorithm of [12]; the main difference is that in our case, we "passively" detect the MIS, by locking on memory variables, and by waiting on preceding ordered threads. This means, that a vertex only "pushes" its cluster ID and status (cluster center/clustered/unclustered) to its friends, versus "pulling" (or asking) for its friends' cluster status. This saves a substantial amount of computational effort.

## 5 Experiments

Our parallel algorithms were all implemented[2] in Scala—we defer a full discussion of the implementation details to Appendix C. We ran all our experiments on Amazon EC2's r3.8xlarge (32 vCPUs, 244Gb memory) instances, using 1-32 threads. The real graphs listed in Table 1 were each

| Graph | # vertices | # edges | Description |
|---|---|---|---|
| DBLP-2011 | 986,324 | 6,707,236 | 2011 DBLP co-authorship network [25, 26, 27]. |
| ENWiki-2013 | 4,206,785 | 101,355,853 | 2013 link graph of English part of Wikipedia [25, 26, 27]. |
| UK-2005 | 39,459,925 | 921,345,078 | 2005 crawl of the .uk domain [25, 26, 27]. |
| IT-2004 | 41,291,594 | 1,135,718,909 | 2004 crawl of the .it domain [25, 26, 27]. |
| WebBase-2001 | 118,142,155 | 1,019,903,190 | 2001 crawl by WebBase crawler [25, 26, 27]. |

Table 1: Graphs used in the evaluation of our parallel algorithms.

tested with 100 different random $\pi$ orderings. We measured the runtimes, speedups (ratio of runtime on 1 thread to runtime on $p$ threads), and objective values obtained by our parallel algorithms. For comparison, we also implemented the algorithm presented in [6], which we denote as CDK for short[3]. Values of $\epsilon = 0.1, 0.5, 0.9$ were used for *C4* BSP, *ClusterWild!* BSP and CDK. In the interest of space, we present only representative plots of our results; full results are given in our appendix.

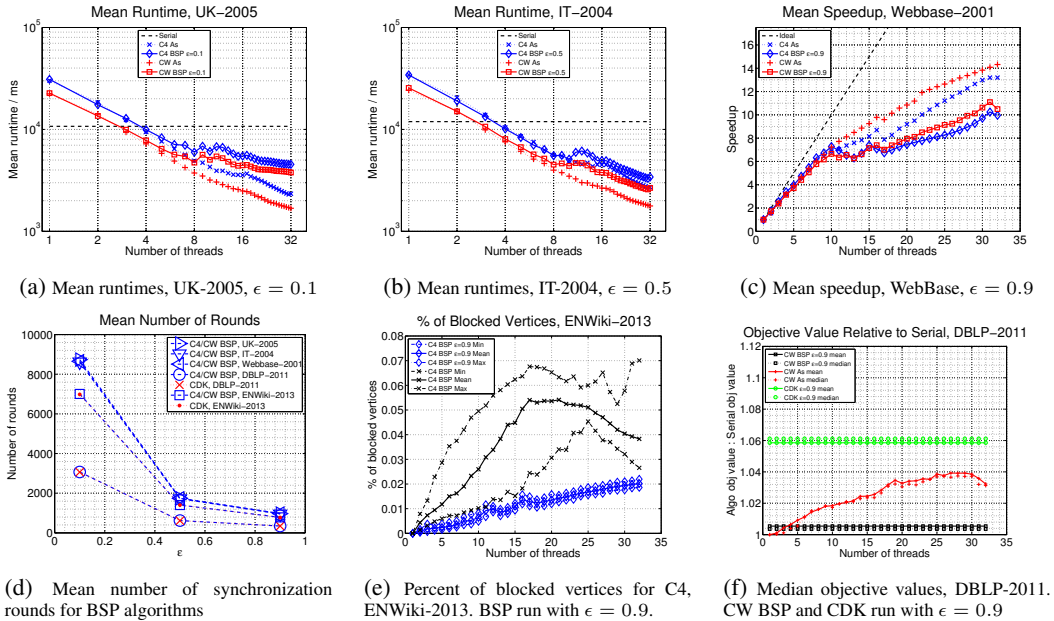

(a) Mean runtimes, UK-2005, $\epsilon = 0.1$

(b) Mean runtimes, IT-2004, $\epsilon = 0.5$

(c) Mean speedup, WebBase, $\epsilon = 0.9$

(d) Mean number of synchronization rounds for BSP algorithms

(e) Percent of blocked vertices for C4, ENWiki-2013. BSP run with $\epsilon = 0.9$.

(f) Median objective values, DBLP-2011. CW BSP and CDK run with $\epsilon = 0.9$

Figure 2: In the above figures, 'CW' is short for *ClusterWild!*, 'BSP' is short for the bulk-synchronous variants of the parallel algorithms, and 'As' is short for the asynchronous variants.

**Runtimes & Speedups:** *C4* and *ClusterWild!* are initially slower than serial, due to the overheads required for atomic operations in the parallel setting. However, all our parallel algorithms outperform *KwikCluster* with 3-4 threads. As more threads are added, the asychronous variants become faster than their BSP counterparts as there are no synchronization barrriers. The difference between BSP and asychronous variants is greater for smaller $\epsilon$. *ClusterWild!* is also always faster than *C4* since there are no coordination overheads. The asynchronous algorithms are able to achieve a speedup of 13-15x on 32 threads. The BSP algorithms have a poorer speedup ratio, but nevertheless achieve 10x speedup with $\epsilon = 0.9$.

**Synchronization rounds:** The main overhead of the BSP algorithms lies in the need for synchronization rounds. As $\epsilon$ increases, the amount of synchronization decreases, and with $\epsilon = 0.9$, our algorithms have less than 1000 synchronization rounds, which is small considering the size of the graphs and our multicore setting.

**Blocked vertices:** Additionally, *C4* incurs an overhead in the number of vertices that are blocked waiting for earlier vertices to complete. We note that this overhead is extremely small in practice— on all graphs, less than 0.2% of vertices are blocked. On the larger and sparser graphs, this drops to less than 0.02% (i.e., 1 in 5000) of vertices.

**Objective value:** By design, the *C4* algorithms also return the same output (and thus objective value) as *KwikCluster*. We find that *ClusterWild!* BSP is at most 1% worse than serial across all graphs and values of $\epsilon$. The behavior of asynchronous *ClusterWild!* worsens as threads are added, reaching 15% worse than serial for one of the graphs. Finally, on the smaller graphs we were able to test CDK on, CDK returns a worse median objective value than both *ClusterWild!* variants.

## 6 Conclusions and Future Directions

In this paper, we have presented two parallel algorithms for correlation clustering with nearly linear speedups and provable approximation ratios. Overall, the two approaches each other support each other—when *C4* is relatively fast relative to *ClusterWild!*, we may prefer *C4* for its guarantees of accuracy, and when *ClusterWild!* is accurate relative to *C4*, we may prefer *ClusterWild!* for its speed.

In the future, we intend to implement our algorithms in the distributed environment, where synchronization and communication often account for the highest cost. Both *C4* and *ClusterWild!* are well-suited for the distributed environment, since they have polylogarithmic number of rounds.

## Footnotes

[1] We skip the constants to simplify the presentation; however they are all smaller than 10.

[2]Code available at `https://github.com/pxinghao/ParallelCorrelationClustering`.

[3]CDK was only tested on the smaller graphs of DBLP-2011 and ENWiki-2013, because CDK was prohibitively slow, often 2-3 orders of magnitude slower than *C4*, *ClusterWild!*, and even *KwikCluster*.

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
