[Supplementary Material · parallelCC_suppl.pdf]

# A Proofs of Theoretical Guarantees

## A.1 Number of rounds for *C4* and *ClusterWild!*

**Lemma 1.** *C4 and ClusterWild! terminate after* $O\left(\frac{1}{\epsilon}\log n \cdot \log \Delta\right)$ *rounds w.h.p.*

*Proof.* We split our proof in two parts.

For *ClusterWild!*, we wish to upper bound the probability

$$q_t = \mathbb{P}\left\{v \text{ not clustered by round } i+t \,\middle|\, \deg_{i+j}(v) \geq \frac{\Delta_i}{2}, 1 \leq j \leq t\right\}.$$

Observe that the above event happens either if no friends of $v$ become activated by round $i + t$, or if $v$ itself does not become activated. Hence, $q_t$ can be upper bounded by the probability that no friends of $v$ become activated by round $i + t$.

In the following, let $d_{i+j}$ denote the degree of vertex $v$ at roudn $i + j$; for simplicity we drop the round indices on $n$ and $P$. The probability, per round, that no friends of $v$ become activated is equal to[4]

$$\frac{\binom{n-d_{i+j}}{P}}{\binom{n}{P}} = \frac{(n-P)!}{(n-P-d_{i+j})!} \cdot \frac{(n-d_{i+j})!}{n!}$$

$$= \frac{\prod_{t=1}^{d_{i+j}} (n - d_{i+1} + t - P)}{\prod_{t=1}^{d_{i+j}} (n - d_{i+1} + t)} = \prod_{t=1}^{d_{i+j}} \frac{n - d_{i+1} + t - P}{n - d_{i+1} + t}$$

$$= \prod_{t=1}^{d_{i+j}} \left(1 - \frac{P}{n - d_{i+1} + t}\right) \leq \left(1 - \frac{P}{n}\right)^{d_{i+j}}$$

$$\leq \left(1 - \frac{\epsilon}{\Delta_i}\right)^{\Delta_i/2} = \left[\left(1 - \frac{\epsilon}{\Delta_i}\right)^{\Delta_i/\epsilon}\right]^{\epsilon/2} \leq e^{-\epsilon/2}.$$

where the last inequality is due to the fact that

$$(1-x)^{1/x} < e^{-1} \text{ for all } x \leq 1.$$

Therefore, the probability of vertex $v$ failing to be clustered after $t$ rounds is at most $q_t \leq e^{-t \cdot \epsilon/2}$. Hence, we have that for any round $i$, the probability that any vertex has degree more than $\Delta_i/2$ after $t$ rounds is at most $n \cdot e^{-t \cdot \epsilon/2}$, due to a simple union bound. If we want that that probability to be smaller than $\delta$, then

$$n \cdot e^{-t \cdot \epsilon/2} < \delta \Leftrightarrow \ln n - t \cdot \epsilon/2 < \ln(\delta) \Leftrightarrow t > \frac{2}{\epsilon} \cdot \ln(n/\delta)$$

Hence, with probability $1 - \delta$, after $\frac{2}{\epsilon} \cdot \ln(n/\delta)$ rounds either all nodes of degree greater than $\Delta/2$ are clustered, or the maximum degree is decreased by half. Applying this argument $\log \Delta$ times yields the result, as the maximum degree of the remaining graph becomes 1.

For *C4* the proof follows simply from the analogous proof of [12]. Consider any round of the algorithm, and break it into $k$ steps (each step, for each vertex in $\mathcal{A}$ that becomes a cluster center). Let $v$ be a vertex that has degree at most $\Delta/2$, and is not active. During step 1 of round 1, the probability that $v$ is not adjacent to $\pi(1)$ is at most $1 - \frac{\epsilon}{2n}$. If $v$ is not selected at step 1, then during step 2 of round 1, the probability that $v$ is not adjacent to the next cluster center is again at most $1 - \frac{\epsilon}{2n}$. After processing all vertices in $\mathcal{A}$, during the first round, either $v$ was clustered, or its degree became strictly less than $\Delta/2$, or the probability that neither of the previous happened is at most $(1 - \frac{\epsilon}{2n})^{\frac{\epsilon\Delta}{n}} \leq 1 - \epsilon/2$. It is easy to see that after $O(\frac{1}{\epsilon}\log n)$ rounds vertex $v$ will have either been clustered or its degree would be smaller than $\Delta/2$. Union bounding for $n$ vertices and all rounds, we get that the max degree of the remaining graph gets halved after $O(\frac{1}{\epsilon}\log n)$ rounds, hence the total number of rounds needed is at most $O(\frac{1}{\epsilon}\log n \log \Delta)$, with high probability.

$\square$

## A.2 Running times

In this section, we prove the running time theorem for our Algorithms. We first present the following recent graph-theoretic result.

**Theorem A.1** (Krivelevich [13]). *Let $G$ be an undirected graph on $n$ vertices, with maximum degree $\Delta$. Let us sample each vertex independently with probability $p = \frac{1-\epsilon}{\Delta}$ and define as $G'$ the induced subgraph on the activated vertices. Then, the largest connected component of the resulting graph $G'$ has size at most $\frac{4}{\epsilon^2} \log n$ with high probability.*

To apply Theorem A.1, we first need to convert it into a result for sampling without replacement (instead of i.i.d. sampling).

**Lemma A.2.** *Let us define two sequences of binary random variables $\{X_i\}_{i=1}^n, \{Y_i\}_{i=1}^n$. The first sequence comprises $n$ i.i.d. Bernoulli random variables with probability $p$, and the second sequence a random subset of $B$ random variables is set to $1$ without replacement, where $B$ is integer that satisfies*

$$(n+1) \cdot p - 1 \leq B < (n+1) \cdot p.$$

*Let us now define $\rho_X = \mathbb{P}\left(f(X_1, \ldots, X_n) > C\right)$ for some $f$ and some number $C$, and similarly define $\rho_Y$ and $\rho_Z$. Let us further assume that we have an upper bound on the above probability $\rho_X \leq \delta$. Then, $\rho_Y \leq n \cdot \delta$.*

*Proof.* By expanding $\rho_X$ using law of total probability we have

$$\rho_X = \sum_{b=0}^n \mathbb{P}\left(f(X_1, \ldots, X_n) > C \,\middle|\, \sum_{i=1}^n X_i = b\right) \cdot \mathbb{P}\left(\sum_{i=1}^n X_i = b\right) = \sum_{b=0}^n q_b \cdot \mathbb{P}\left(\sum_{i=1}^n X_i = b\right)$$

where $q_b$ is the probability that $f(X_1, \ldots, X_n) > C$ given that a uniformly random subset of $b$ variables was set to $1$. Moreover, we have

$$\rho_Y = \sum_{b=0}^n \mathbb{P}\left(f(Y_1, \ldots, Y_n) > C \,\middle|\, \sum_{i=1}^n Y_i = b\right) \cdot \mathbb{P}\left(\sum_{i=1}^n Y_i = b\right) \overset{(i)}{=} \sum_{b=0}^n q_b \cdot \mathbb{P}\left(\sum_{i=1}^n Y_i = b\right)$$

$$\overset{(ii)}{=} q_B \cdot 1 \tag{1}$$

where $(i)$ comes form the fact that $\mathbb{P}\left(f(Y_1, \ldots, Y_n) > C \,|\, \sum_{i=1}^n Y_i = b\right)$ is the same as the probability that that $f(X_1, \ldots, X_n) > C$ given that a uniformly random subset of $b$ variables where set to $1$, and $(ii)$ comes from the fact that since we sample without replacement in $Y$, we have that $\sum_i^n Y_i = B$ always.

If we just keep the $b = B$ term in the expansion of $\rho_X$ we get

$$\rho_X = \sum_{b=0}^n q_b \cdot \mathbb{P}\left(\sum_{i=1}^n X_i = b\right) \geq q_B \cdot \mathbb{P}\left(\sum_{i=1}^n X_i = B\right) = \rho_Y \cdot \mathbb{P}\left(\sum_{i=1}^n X_i = B\right) \tag{2}$$

since all terms in the sum are non-negative numbers. Moreover, since $X_i$s are Bernoulli random variables, then $\sum_{i=1}^n X_i$ is Binomially distributed with parameters $n$ and $p$. We know that the maximum of the Binomial pmf with parameters $n$ and $p$ occurs at $\mathbb{P}\left(\sum_i X_i = B\right)$ where $B$ is the integer that satisfies $(n+1) \cdot p - 1 \leq B < (n+1) \cdot p$. Furthermore we know that the maximum value of the Binomial pmf cannot be less than $\frac{1}{n}$, that is

$$\mathbb{P}\left(\sum_{i=1}^n X_i = B\right) \geq \frac{1}{n}. \tag{3}$$

If we combine (2) and (3) we get $\rho_X \geq \rho_Y/n \Leftrightarrow \rho_Y \leq n \cdot \delta$. □

**Corollary A.3.** *Let $G$ be an undirected graph on $n$ vertices, with maximum degree $\Delta$. Let us sample $\epsilon \cdot \frac{n}{\Delta}$ vertices without replacement, and define as $G'$ the induced subgraph on the activated vertices. Then, the largest connected component of the resulting graph $G'$ has size at most $\frac{4}{\epsilon^2} \log n$ with high probability.*

We use this in the proof of our theorem that follows.

**Theorem 2.** *The theoretical running time of C4 on $P$ cores is upper bounded by $O\left(\left(\frac{m+n\log n}{P}+P\right)\log n \cdot \log \Delta\right)$ as long as the number of cores $P$ is smaller than $\min_i \frac{n_i}{\Delta_i}$, where $\frac{n_i}{\Delta_i}$ is the size of the batch in the $i$-th round of each algorithm. The running time of Cluster-Wild! on $P$ cores is upper bounded by $O\left(\left(\frac{m+n}{P}+P\right)\log n \cdot \log \Delta\right)$.*

*Proof.* We start with analyzing *C4*, as the running time of *ClusterWild!* follows from a similar, and simpler analysis. Observe, that we operate on Bulk Synchronous Parallel model: we sample a batch of vertices, $P$ cores asynchronously process the vertices in the batch, and once the batch is empty there is a bulk synchronization step. The computational effort spent by *C4* can be split in three parts: i) computing the maximum degree, ii) creating the clusters, per batch, iii) syncronizing at the end of each batch.

**Computing $\Delta$ and synhronizing cost.** Computing $\Delta_i$ at the begining of each batch, can be implemented in time $\frac{m_i}{P} + \log P$, where each thread picks $n_i/P$ vertices and computes locally their degrees, and inserts it to a sorted data structure (*e.g.*, a B-tree that admits parallel operations), and then we get the largest item in logarithmic time. Moreover, the third part of the computation, i.e., synchronization among cores, can be done in $O(P)$. A little more involved argument is needed for establishing the running time of the second part, where the algorithms create the clusters.

**Clustering cost.** For a single vertex $v$ sampled by a thread, the time required by the thread to process that vertex is the sum of the time needed to 1) wait inside the attemptCluster for preceding friends (by the order of $\pi$), 2) "send" its $\pi(v)$ to its friends, if $v$ is a cluster center. 3) if $v$ is a cluster center, then for each $u$ friends it will attempt to update clusterID$(u)$; however, this thread potentially competes with other threads that are attempting to write in clusterID$(u)$ at the same time.

Using Corollary A.3, we can show that no more than $O(\log n)$ threads compete with each other at the same time, with high probability. Observe, that in our sampling scheme of batches of vertices, we are taking the first $B_i = \frac{\epsilon}{\Delta_i} \cdot n$ elements of a random prefix $\pi$. This is equivalent to sampling $B_i$ vertices without replacement from the graph $G_i$ of the current round. A slight modification of Theorem A.1, gives us that the largest connected component in the sampled subgraph is at most $O(\log n)$, with high probability. This directly implies that a thread cannot be waiting for more than $O(\log n)$ other threads inside `attemptCluster`$(v)$. Therefore, the time spent by each thread to wait on other threads in `attemptCluster`$(v)$ is upper bounded by the number of maximum threads that it can be friends with (which is at most $O(\log n)$, times the time it takes each of these threads to be done with their execution, which is at most $\Delta_i \log n$ (even assuming the worst case conflict pattern when updating at most $\Delta_i$ entries in the clusterID array). Hence, for *C4* the processing time of a single vertex is upperbounded by $O(\Delta_i \cdot \log^2 n)$.

**Job allocation.** Now, observe that when each thread is done processing vertex, it picks the next vertex from $\mathcal{A}$ (if $\mathcal{A}$ is not empty). This process essentially models a classical greedy task assignment to cores, that leads to a 2 approximation, where the optimum allocation leads to a max weight among cores that is at least equal to $\max(\Delta_i, B_i\Delta_i/P)$. This implies that the running time on $P$ asynchronous threads of a single batch, is upperbounded by

$$O\left(max\left(\Delta_i \log n, \frac{B_i\Delta_i \log^2 n}{P}\right)\right) = O\left(max\left(\Delta_i \log n, \frac{n_i \log^2 n}{P}\right)\right).$$

Assuming, that the number of cores, is always less than the batch size (a reasonable assumption, as more cores, would not lead to further benefits), we obtain that the time for a single batch is

$$O\left(\frac{E_i}{P} + \frac{n_i \log^2 n}{P} + P\right).$$

Observe that a difference in *ClusterWild!*, is that waiting is avoided, hence, the running time, per batch of *ClusterWild!* is

$$O\left(\frac{E_i}{P} + \frac{n_i}{P} + P\right).$$

Multiplying the above, with the number of rounds given by Lemma 1, we obtain the theorem.

□

### A.3 Approximation Guarantees

One can view the execution of ClusterWild! on $G$ as having *KwikCluster* run on a "noisy version" of $G$. A main issues is that *KwikCluster* never allows two friends in the original graph to become cluster centers. Hence, since ClusterWild! ignores these edges among active vertices, one can view these edges as "adverserially" deleted. The major technical contribution of this work is to quantify how these "ignored" edges affect the quality of the output solution.

Before we proceed, let us define the bad combinatorial structures are charged for the cost of our solution.

**Definition 2.** *Let us define as a bad triangle, a set of three vertices in $G$ such that two pairs are joined with a positive edge and one pair is joined with a negative edge.*

The following simple lemma is useful in quantifying the cost of the output clustering for *any* peeling algorithm.

**Lemma 5.** *The cost of any greedy algorithm that picks a vertex $v$ (irrespective of the sampling order), creates $\mathcal{C}_v$, peels it away and repeats, is equal to the number of bad triangles adjacent to each cluster center $v$.*

*Proof.* Consider the first step of the algorithm , for simplicity, and without loss of generality. Let us define as $T_{\text{in}}$ the number of vertex pairs inside $\mathcal{C}_v$ that are not friends (i.e., they are joined by a negative edge). Moreover, let $T_{\text{out}}$ denote the number of vertices outside $\mathcal{C}_v$ that are friends with vertices inside $\mathcal{C}_v$. Then, the number of disagreements (i..e, number of misplaced pairs of vertices) generated by cluster $\mathcal{C}_v$, is equal to $T_{\text{in}} + T_{\text{out}}$.

Observe that all the $T_{\text{in}}$ edges are negative, and all $T_{\text{out}}$ are positive ones. Let for example $(u, w)$ be one of the $T_{\text{in}}$ negative edges inside $\mathcal{C}_v$, hence both $u, w$ belong to $\mathcal{C}_v$ (i.e., are friends with $v$). Then, $(u, v, w)$ forms a *bad triangle*. Similarly, for every edge that is incident to a vertex in $\mathcal{C}_v$, with one end point say $u' \in \mathcal{C}_v$ and one $w' \in V \backslash v$, the triangle formed by $(v, u', w')$, is also a bad triangle.

Hence, all edges that are accounted for in the final cost of the algorithm (i..e, total number of disagreements) are equal to the $T_{\text{in}} + T_{\text{out}}$ bad triangles that include these edges and each cluster center per round. □

Let us now consider the set of all cluster centers generated by ClusterWild!; call these vertices $\mathcal{C}_{\text{CW}}$. Then, consider the graph $G'$ that is generated by deleting all edges between $\mathcal{C}_{\text{CW}}$. Observe that this is a random graph, since the set of edges deleted depends on the specific random sampling that is performed in ClusterWild!. Let $\mathcal{E}_{del}$ denote the set of edges deleted in this new graph $G'$. We will use the following simple technical proposition to quantify how many more bad triangles $G'$ has compared to $G$.

**Proposition A.4.** *Given any graph $G$ with positive and negative edges, then let us obtain a graph $G_e$ where we have removed a single edge, $e$ from $G$. Then, the $G_e$ has at most $\Delta$ more bad triangles compared to $G$.*

*Proof.* Let $(i, j, k)$ be a bad triangle in $G$ but not in $G_e$. Then it must be the case that $e \in t$. WLOG let $e = (i, j)$, and so $k \in N(i) \cup N(j)$. Since $|N(i) \cup N(j)| \leq min(deg_i, deg_j) \leq \Delta$, there can be at most $\Delta$ new bad triangles in $G_e$. □

Now, assume a random permutation $\pi$ for which we run *ClusterWild!*, and let $\hat{\mathcal{A}} = \cup_{r=1}^{R} \mathcal{A}_r$ denote the union of all active sets of vertices, for each round $r$ of the algorithm. Moreover, let $\hat{G}$, denote the graph that is missing all edges between the vertices in the sets $\mathcal{A}_r$. A simple way to bound the clustering error of *ClusterWild!*, is splitting it in to two terms: the number of old bad triangles of $G$ adjacent to active vertices, plus the number of *all* new triangles induced by ignoring edges. Observe that this bound can be loose, since not all new bad triangles of $\hat{G}$ count towards the clustering error. However, this makes the analysis tractable.

Lemma 6 then follows.

**Lemma 6.** *Let $\hat{G}$ denote the random graph induced by deleting all edges between active vertices per round, for a given run of ClusterWild!, and let $\tau_{new}$ denote the number of additional bad triangles that $\hat{G}$ has compared to $G$. Then, the expected cost of ClusterWild! can be upper bounded as $\mathbb{E}\left\{\sum_{t \in \mathcal{T}_b} \mathbf{1}_{\mathcal{P}_t} + \tau_{new}\right\}$, where $\mathcal{P}_t$ is the event that triangle $t$, with end points $i, j, k$, is bad, and at least one of its end points becomes active, while $t$ is still part of the original unclustered graph.*

# B Implementation Details

A number of implementation tricks were needed to make our parallel algorithms scalable in practice. We discuss these details and any deviation from the theory in this section.

## B.1 Atomic and non-atomic variables in Java/Scala.

In Java/Scala, processors maintain their own local cache of variable values, which could lead to spinlocks in *C4* or greater errors in *ClusterWild!*. It is necessary to enforce a consistent view across all processors by the use of synchronization or AtomicReferences, but doing so will incur high overheads that render the algorithm unscalable.

To mitigate this overhead, we exploit a monoticity property of our algorithms—the clusterID of any vertex is a non-increasing value. Thus, many of the checks in *C4* and *ClusterWild!* may be sufficiently performed using only an outdated version of clusterID. Hence, we may maintain both an inconsistent but cheap clusterID array as well as an expensive but consistent atomic clusterID array. Most reads can be done using the cheap inconsistent array, but writes must propagate to the consistent atomic array. Since each clusterID is written a few times but read often, this allows us to minimize the cost of synchronizing values without any substantial changes to the algorithm itself.

We point out that the same concepts may be applied in a distributed setting to minimize communication costs.

## B.2 Estimating but not computing $\Delta$.

As written, the BSP variants require a computation of the maximum degree $\Delta$ at each round. Since this effectively involves a scan of all the edges, it can be an expensive operation to perform at each iteration. We instead use a proxy $\hat{\Delta}$ which is initialized to $\Delta$ in the first round, and halved every $\frac{2}{\epsilon} \ln(n \log \Delta/\delta)$ rounds.

With a simple modification to Lemma 1, we can see that w.h.p. any vertex with degree greater than $\hat{\Delta}$ will either be clustered or have its degree halved after $\frac{2}{\epsilon} \ln(n \log \Delta/\delta)$ rounds, so $\hat{\Delta}$ upper-bounds $\Delta$ and our algorithms complete in logarithmic number of rounds.

## B.3 Lazy deletion of vertices and edges.

In practice, we do not remove vertices and edges as they are clustered, but simply skip over them when they are encountered later in the process. We find that this approach decreases the runtimes and overall complexity of the algorithm. (In particular, edges between spokes may never be touched in the lazy deletion scheme, but must nevertheless be removed in the proactive deletion approach.) Lazy deletions also allow us to avoid expensive mutations of internal data structures.

## B.4 Binomial sampling instead of fixed-size batches

Lazy deletion does introduce an extra complication, namely it is now more difficult to sample a fixed-size batch of $n_i \epsilon/\Delta$ vertices, where $n_i$ is the number of remaining unclustered vertices. This is because we do not maintain a separate set of $n_i$ unclustered vertices, nor explicitly compute the value of $n_i$.

We do, however, maintain a set of *unprocessed* vertices, that is, a suffix of $\pi$ containing $n_i$ unclustered vertices and $m_i$ clustered vertices that have not been passed through by the algorithm. We may therefore resort to an i.i.d. sampling of these vertices, choosing each with probability $\epsilon/\Delta$. Since processing an unprocessed but clustered vertex has no effect, we effectively simulate an i.i.d. sampling of the $n_i$ unclustered vertices.

Furthermore, we do not have to actually sample each vertex—because $\pi$ is a uniform random permutation, it suffices to draw $B \sim Bin(n_i + m_i, \epsilon/\Delta)$ and extract the next $B$ elements from $\pi$ for processing, reducing the number of random draws from $n_i + m_i$ Bernoullis to a single Binomial.

All of our theorems hold in expectation when using i.i.d. sampling instead of fixed-size batches.

## B.5 Comment on CDK Implementation

A crucial difference between the CDK algorithm and our algorithms lies in the fact that CDK might reject vertices from the active set, which are then placed back into the set of unclustered vertices for potential selection at later rounds. Conversely, our algorithms ensure that the active set is always completely processed, so any vertex that has been selected will no longer be selected in an active set again. We are therefore able to exploit a single random permutation $\pi$ and use the tricks with lazy deletions and binomial sampling that are not available to CDK, which instead has to perform the complete i.i.d. sampling. In our opinion, this accounts for the largest difference in runtimes between CDK and our algorithms.

# C Full experiment results

(a) UK-2005, $\epsilon = 0.1$

(b) UK-2005, $\epsilon = 0.5$

(c) UK-2005, $\epsilon = 0.9$

(d) IT-2004, $\epsilon = 0.1$

(e) IT-2004, $\epsilon = 0.5$

(f) IT-2004, $\epsilon = 0.9$

(g) Webbase-2001, $\epsilon = 0.1$

(h) Webbase-2001, $\epsilon = 0.5$

(i) Webbase-2001, $\epsilon = 0.9$

(j) ENWiki-2013, $\epsilon = 0.1$

(k) ENWiki-2013, $\epsilon = 0.5$

(l) ENWiki-2013, $\epsilon = 0.9$

(m) DBLP-2011, $\epsilon = 0.1$

(n) DBLP-2011, $\epsilon = 0.5$

(o) DBLP-2011, $\epsilon = 0.9$

**Figure 3:** Empirical mean runtimes. For short, 'CW' is *ClusterWild!* and 'As' refers to the asynchronous variants. On larger graphs, our parallel algorithms on 3-4 threads are faster than serial *KwikCluster*. On the smaller graphs, the BSP variants have expensive synchronization barriers (relative to the small amount of actual work done) and do not necessary run faster than serial *KwikCluster*; the asynchronous variants do outperform serial *KwikCluster* with 4-5 threads. We were only able to run CDK on the smaller graphs, for which CDK was 2-3 orders of magnitude slower than serial. Note also that the BSP variants have improved runtimes for larger $\epsilon$.

**Figure 4:** Empirical mean speedups. The best speedups (14x on large graphs) are achieved by asynchronous *ClusterWild!* which has the least coordination, followed by asynchronous *C4* (13x on large graphs). The BSP variants achieve up to 10x speedups on large graphs, with better speedups as $\epsilon$ increases. On small graphs we obtain poorer speedups as the cost of any contention is magnified as the actual work done is comparatively small. There are a couple of kinks at 10 and 16 threads, which we postulate is due to NUMA and hyperthreading effects—the EC2 r3.8xlarge instances are equipped with 10-core Intel Xeon E5-2670 v2 (Ivy Bridge) processors with 32 vCPUs and hyperthreading.

(a) UK-2005, $\epsilon = 0.1$      (b) UK-2005, $\epsilon = 0.5$      (c) UK-2005, $\epsilon = 0.9$

(d) IT-2004, $\epsilon = 0.1$      (e) IT-2004, $\epsilon = 0.5$      (f) IT-2004, $\epsilon = 0.9$

(g) Webbase-2001, $\epsilon = 0.1$      (h) Webbase-2001, $\epsilon = 0.5$      (i) Webbase-2001, $\epsilon = 0.9$

(j) ENWiki-2013, $\epsilon = 0.1$      (k) ENWiki-2013, $\epsilon = 0.5$      (l) ENWiki-2013, $\epsilon = 0.9$

(m) DBLP-2011, $\epsilon = 0.1$      (n) DBLP-2011, $\epsilon = 0.5$      (o) DBLP-2011, $\epsilon = 0.9$

Figure 5: Empirical objective values relative to mean objective value obtained by serial algorithm.

(a) UK-2005, $\epsilon = 0.1$

(b) UK-2005, $\epsilon = 0.5$

(c) UK-2005, $\epsilon = 0.9$

(d) IT-2004, $\epsilon = 0.1$

(e) IT-2004, $\epsilon = 0.5$

(f) IT-2004, $\epsilon = 0.9$

(g) Webbase-2001, $\epsilon = 0.1$

(h) Webbase-2001, $\epsilon = 0.5$

(i) Webbase-2001, $\epsilon = 0.9$

(j) ENWiki-2013, $\epsilon = 0.1$

(k) ENWiki-2013, $\epsilon = 0.5$

(l) ENWiki-2013, $\epsilon = 0.9$

(m) DBLP-2011, $\epsilon = 0.1$

(n) DBLP-2011, $\epsilon = 0.5$

(o) DBLP-2011, $\epsilon = 0.9$

Figure 6: Empirical percentage of blocked vertices. Generally the number of blocked vertices increases with the number of threads and larger $\epsilon$ values. *C4* BSP has fewer blocked vertices than asynchronous *C4*, but at the cost of more synchronization barriers. We point out that across all 100 runs of every graphs, the maximum percentage of blocked vertices is less than 0.25%; for large sparse graphs, the maximum percentage is less than 0.025%, i.e., 1 in 4000.

## Footnotes

[4]This follows from a simple calculation on the pdf of the hypergeometric distribution.