[Reviews · NeurIPS 2015]

Submitted by Assigned_Reviewer_1

Summary

This work addresses an important special case of the correlation clustering problem: Given as input a graph with edges labeled -1 (disagreement) or +1 (agreement), the goal is to decompose the graph so as to maximize agreement within components. Building on recent work [5,6], this paper contributes two concurrent algorithms, a proof of their approximation ratio, a run-time analysis as well as a set of experiments which demonstrate convincingly the advantage of the proposed algorithms over the state of the art.

Quality

This paper is of high quality. It makes a clear theoretical contribution that is relevant in important ML applications.

Clarity

The paper is written clearly. A strong effort has been made by the authors to make it easy for the reader to follow the formal discussion. While this approach to writing works for this paper and might help to attract more readers, I would have preferred a more formal style, also in the main document.

It is true that the term correlation clustering was introduced in [14]. I would also agree that [14] was the work that popularized the problem in the ML community. At the same time, I would like to see even earlier work referenced, e.g.: Chopra, Sunil and Rao, M.R. The partition problem. Mathematical Programming 59(1-3) pages 87-115. Springer 1993

Originality

The algorithms proposed in this paper are an original contribution. The trade-off between run-time complexity and approximation ratio they achieve (which is established in this paper) is different from that of related algorithms.

Significance

Correlation clustering is a central problem in machine learning and this paper makes a strong contribution to the solution of instances of this problem that arise in practical applications. The work presented here builds on [3,10] and the more recent work [5,6] that clearly deserves the attention form the ML community.
Summary: This high-quality paper contributes two concurrent algorithms for an important clustering problem (correlation clustering with edge weights +1 and -1). It establishes for these algorithms a run-time complexity and approximation ratio by which they compare favorably to the state of the art, which is demonstrated also experimentally.

Submitted by Assigned_Reviewer_2

Two new algorithms are introduced for parallel correlation clustering. The authors demonstrate that each algorithm has a proven ability to outperform serial methods in terms of runtimes. In practice this advantage is recognized most strikingly when using a high number of threads (the algorithmic overhead can show small performance slowdowns because of overhead). Both of the new algorithms show provably good approximation ratios, where one guarantees a 3-approximation ratio and the other comes close by abandoning consistency.

The paper itself is clearly written. It is well outlined and each section is clear to its purpose and results. The theoretical guarantees seem sound, and are the main highlight of the paper. Both algorithms seem to be incremental in their advances past the original KwikCluster algorithm and the distributed versions outlined in [5] and [6], though the speedup they describe is not incremental.

The advances the authors detail are significant in their advances. However, the impact is less clear. In their last paragraphs the authors mention the possible challenges of working with their algorithms in a distributed environment (mainly I/O costs). Given that these environments are becoming very popular for hosting the types of big datasets that these algorithms are designed to cluster, it would have been nice to learn more about typical runtimes in distributed systems, or if there are other types of speedups that could be leveraged.
Summary: The authors clearly present two new enhancements tot he KwikCluster algorithm that allow for parallelization, showing great speedups with provably good approximation.

Submitted by Assigned_Reviewer_3

-Assuming 'kwikcluster' is the current state-of-the-art (this is the only comparison provided in the paper), then the two new algorithms do provide improved speed and comparable accuracy for clustering up to billion-edge graphs (as demonstrated in experimentation on several different datasets) -The technical explanation in the paper could be better written -Seems interesting to those in the correlation clustering field - could have application on large datasets. -Originality - seems low.
Summary: Presentation of two new correlation clustering algorithms, suitable for use on big graphs.

Submitted by Assigned_Reviewer_4

The authors present a parallel version of a classic approximated solution to the Correlation Clustering problem on complete unweighted signed graphs. They prove it preserves the 3 approximation ratio while requiring less peeling rounds and allowing nearly linear speed-up. Relaxing synchronization constraints, they also introduce a variant which improve running time at the cost of introducing additional error compared with the serial solution. These claims are supported by experiments on real world data.

In my opinion, this paper addresses a major problem and provide a solution which is simple, but also well founded and efficient as demonstrated both theoretically and practically. Since its inception in 2002, Correlation Clustering has drawn a lot of attention. Yet until recently, most of the efforts have been directed to either improve the approximation ratio or devise heuristics in a serial fashion. This has started to change in the last years [1a,2a,3a] yet the algorithms proposed here compared favourably with the state of the art. Moreover the exposition is clear and the paper well written.

On the other hand, a few points could be improved.

- First, the main hypothesis (i.e. that the graph is complete) should

be more visible. For instance, the title could replace "Big" by

"Complete" to be more accurate and descriptive. It should also

appear in the abstract, to avoid confusing the reader in a

rush. Furthermore, the graphs presented in the experimental sections

are clearly not complete. Therefore it should be explained how this

was accommodated (for example, considering all missing edges to be

negative yield the same practical result than if this was actually

the case, but it lower the theoretical approximation ratio from

O(log n) to 3 and thus must be carefully justified). Also, the view

of ClusterWild! as a procedure on a noisy graph with deleted

edges has to be clarified in that perspective of complete graphs

(are they transformed into negative edges).

- Second the proof of the running time depends crucially of a recent

result to upper bound the number of competing threads. This result

has not yet been peer-reviewed.

- Third, the sketch of the proof of theorem 4 is a bit confusing to

me. Namely it is not clear how ignoring edges induces *new* bad

triangles in \hat{G}

Minor comments: line 117: C_i inter C_j should be equal to the empty set line 124: missing point after clustered line 136: write iff in full line 169, right: clusterID(v) not u line 173: I assumed Gamma(v) is the positive neighbours of v ordered by pi but it's not defined line 221: define speed-up here rather than in line 371 line 248: it's not clear where in the appendix are the full details line 269: should it be intersection? line 273: I assumed p_t is the probability of the event \mathcal{P}_t from line 280: do \mathcal{S}_u,v S and S_u,v refer to the same quantity? line 285: define C_u,v line 296: is "positive" needed to qualify neighbours? It doesn't seem to appear in the following line line 299: shouldn't the outer be P/n line 301: maybe remind that P<\epsilon n/\delta. Is it by the way? Theorem 2 doesn't mention epsilon line 308: is \delta the same as in line 508? line 308: 4 \epsilon / (1 - exp(-4\epsilon)) is equivalent to

1+2\epsilon around 0, giving a factor of (3+4\epsilon) to OPT. Yet the

experiments use \epsilon=0.9, in which case

1 + 2 * [4 \epsilon /

(1 - exp(-4\epsilon))]  8.4 instead of 3. line 360: appendix B line 365: it's not clear to me what is the semantic of signed edges in that case Figure 2: the legends are barely readable when printed line 427: is there any way to tell which version to prefer without running both of them? line 431: could local heuristics of [3] be parallelized as well? line 461: [17] was accepted in COLT'15 line 500: typo in round line 693: should it be intersection?

[1a] Ahn, K., Cormode, G., Guha, S., McGregor, A., & Wirth, A. (2015). Correlation Clustering in Data Streams. In Proceedings of The 32nd International Conference on Machine Learning (pp. 2237-2246). Retrieved from http://jmlr.org/proceedings/papers/v37/ahn15.html [2a] Bonchi, F., Gionis, A., & Ukkonen, A. (2012). Overlapping correlation clustering. Knowledge and Information Systems, 35(1), 1-32. doi:10.1007/s10115-012-0522-9 [3a] Chierichetti, F., Dalvi, N., & Kumar, R. (2014). Correlation clustering in MapReduce. In Proceedings of the 20th ACM SIGKDD international conference on Knowledge discovery and data mining - KDD '14 (pp. 641-650). New York, New York, USA. doi:10.1145/2623330.2623743

Summary: This paper proposes two variants for parallel correlation clustering. The paper is interesting and well written.

Author Feedback
Author rebuttal: We would like to thank the Reviewers for the positive assessment of our work. We are glad that they enjoyed reading our paper.
Most of the comments raised consider some technical clarifications and stylistic details.
We provide detailed responses below.

===================
Assigned_Reviewer_1
We thank the Reviewer for the positive feedback.

1. "... I would have preferred a more formal style ..."
The paper was deliberately written for accessibility to the general audience. We think that our formal proofs would be easier to follow once the intuition was made clear in the main text.
We tried to cater to the more technical audience in our Appendix.

2. "... I would like to see even earlier work referenced ..."
Thank you for pointing out the reference of Chopra, Sunil and Rao.
We will add a detailed reference in the final version of the paper.

===================
Assigned_Reviewer_2
We thank the Reviewer for the positive feedback.

1. "... However, the impact is less clear. ..."
As other reviewers have noted, we present some of the first approaches to scale up correlation clustering, both provably and in experiments.
Our proposed algorithms come with provable guarantees on runtime and performance, and extensive experimental evaluation.
Our parallel CC algorithms significantly outperform the serial algorithm, and the state-of-the-art distributed algorithm, which is important when dealing with increasingly large graphs.

2. "... it would be nice to learn more about typical runtimes in distributed systems ..."
We share the Reviewer's interest in applying the algorithms to the distributed setting. However, this was not the focus of the paper.
The implementation and discussion of distributed version of the algorithms were left as future work.

===================
Assigned_Reviewer_3
We thank the Reviewer for the positive review and detailed comments.

1. "... First, the main hypothesis (i.e. that the graph is complete) should be more visible ..."
We will make this clear in the final version of the paper.
In the experiments we assumed the graph to be complete: the absence of an edge was assumed to imply a negative edge.

2. "...the view of ClusterWild! as a procedure on a noisy graph with deleted edges has to be clarified in that perspective of complete graphs."
When referring to "deletion" of an edge in ClusterWild!, we meant that the (positive) edge is flipped to a negative.

3. "... the proof of the running time depends crucially on a recent result ..."
The result of [Krivelevich 2014] has an elegant and simple proof which we checked, and is easy to follow.

4. "... Namely it is not clear how ignoring edges induces *new* bad triangles in \hat{G}."
A bad triangle is defined as a 3-clique, where one of the edges is negative. If we had a triangle where all edges are positive, then flipping one to negative, would instantly create 1 new bad triangle.
This generalizes: a single flipped edge can introduce at most O(maxDegree) new bad triangles.

5. We are grateful for the numerous typos and errors pointed out by the Reviewer. We will address all of them in the final version.

===================
Assigned_Reviewer_4
We thank the Reviewer for the overall positive review.

We would like to emphasize again our contributions over previous work
- Our parallel algorithms are demonstrably faster than the state-of-the-art.
- The analysis of the runtimes of our algorithms is novel, based on a recent graph-theoretic result.
- We proposed a novel approach of viewing the coordination-free algorithm (ClusterWild!) as operating on a noisy graph. This allowed us to build on prior work and analyze the approximation guarantees of ClusterWild!

===================
Assigned_Reviewer_5
We thank the Reviewer for the positive review.

The Reviewer commented that the presentation of the paper could be further improved.
All stylistic issues and typos will be addressed per all the Reviewers' requests.

===================
Assigned_Reviewer_6
We thank the Reviewer for the feedback.

1. "... The technical explanation in the paper could be better written ..."
We made an effort to offer a high-level description of our technical results in the main text of the paper.
To do this we delegated all technical details for the Appendix.

2. "... Originality - seems low ..."
We refer the Reviewer to our reply to Assigned_Reviewer_4, as well as the reviews by other reviewers.
The algorithmic innovations we made are what enabled us to significantly outperform KwikCluster and the state-of-the-art distributed algorithm.
Further, our run-time and approximation analyses are novel, and based on recent graph-theoretic developments.